# Physical and Biological Treatment Technologies of Slaughterhouse Wastewater: A Review

**Mohammed Ali Musa** [1,2] **and Syazwani Idrus** [1,*]

1   Department of Civil Engineering, Faculty of Engineering, Universiti Putra Malaysia,
    Serdang 43400, Selangor, Malaysia; alisulezee@gmail.com
2   Department of Civil and Water Resources Engineering, University of Maiduguri,
    Maiduguri P.M.B. 1069, Borno State, Nigeria
*   Correspondence: syazwani@upm.edu.my; Tel.: +60-13-692-2301

**Abstract:** Physical and biological treatment technology are considered a highly feasible and economic way to treat slaughterhouse wastewater. To achieve the desired effluent quality for disposal or reuse, various technological options were reviewed. However, most practical operations are accompanied by several advantages and disadvantages. Nevertheless, due to the presence of biodegradable organic matter in slaughterhouse waste, anaerobic digestion technology is commonly applied for economic gain. In this paper, the common technologies used for slaughterhouse wastewater treatment and their suitability were reviewed. The advantages and disadvantages of the different processes were evaluated. Physical treatments (dissolved air floatation (DAF), coagulation–flocculation and sedimentation, electrocoagulation process and membrane technology) were found to be more effective but required a large space to operate and intensive capital investment. However, some biological treatments such as anaerobic, facultative lagoons, activated sludge process and trickling filters were also effective but required longer start-up periods. This review further explores the various strategies being used in the treatment of other wastewater for the production of valuable by-products through anaerobic digestion.

**Keywords:** wastewater; slaughterhouse wastewater; physical treatment; biological treatment

## 1. Introduction

The effective treatment of high-strength industrial wastewater has increased over time due to the effects related to environmental pollution. The discharge of untreated slaughterhouse wastewater (SWW) constitutes a severe threat to public health and the environment [1]. Although rivers have a natural cleansing capacity, the frequent release of such effluent without it being adequately treated first might overburden the receiving water body. Today, the management of wastewater needs to incorporate both waste minimization and resource recovery [2]. Although fresh water consumption by different slaughterhouse industry varies in magnitude and concentration, it is usually preferable to minimize wastewater generation at its source. The wastewater generated from a slaughterhouse consists of organic by-products which are considered industrial organic wastes, which are challenging to treat due to their high protein and lipid contents. The main organic streams that portrayed SWW as recalcitrant in nature include the blood, paunches from the removal of the rumen and the intestinal content, the intestinal residues from the evisceration process [3], fats from the meat trim step as well as the head and the limbs (mostly bones). Conventionally, SWW treatment methods are similar to the current technologies used in municipal wastewater treatment, which include physicochemical and biological treatments where each method has its advantages and disadvantages.

### 1.1. Physicochemical Treatment

Physicochemical treatment methods usually involve solid separation from the fluid. It is recommended that the effluent be sent for primary or secondary treatment after the preliminary treatment depending on the intensity of the SWW [4]. Dissolved air floatation (DAF), coagulation–flocculation and sedimentation, electrocoagulation process and membrane technology are usually employed as primary treatment technologies for the treatment of SWW [5,6]. In the analysis of samples, the standard methods for the examination of water and wastewater of the American Public Health Association [7] are commonly applied, to achieve chemical oxygen demand (COD), biochemical oxygen demand (BOD), total suspended solid (TSS), volatile suspended solids (VSS), ammonia nitrogen ($NH_3$-N), fats, oil and grease (FOG), colour and turbidity removals.

#### 1.1.1. Dissolved Air Floatation (DAF)

Dissolved air floatation is simply the introduction of air from the bottom of the system for liquid–solid separation, as shown in Figure 1. During operation, the FOG light solid materials are transported to the surface, creating a sludge blanket. Thus, the scum formed is continuously removed by scrapping.

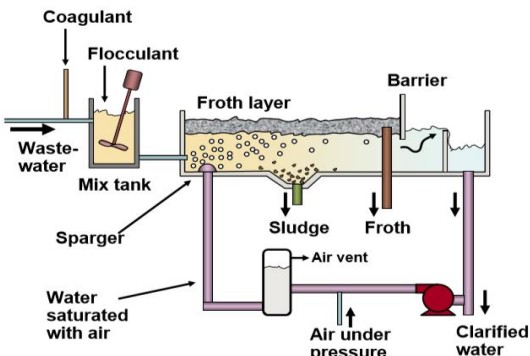

**Figure 1.** Schematic diagram of the dissolved air floatation clarifier unit [8].

Polymers and other flocculants are usually applied to enhance the efficiency of DAF. In treating SWW, ferric chloride and aluminium sulphate are usually added to facilitate the aggregation and precipitation of protein in addition to fat and grease floatation. Moreover, 30 to 90% COD, as well as 70 to 80% BOD removal efficiency can be achieved using the DAF process. Furthermore, Mittal. [9] and De Nardi et al. [5] have shown that the DAF system is capable of achieving moderate to high nutrient removal. Floatation can also be used as an alternative method of handling pulp and paper mill effluent in addition to firm settling. These devices inject a pressurized flow of air-saturated water at the base of a deep chest that holds the paper mill process steam.

The injected water s released into the chest, and tiny air bubbles come out of the solution and start to rise. The rising bubbles tend to carry any other fairly solid binding particles and can easily be skimmed from the water's surface. DAF's main drawback, however, is commonly associated with relatively frequent system failure and inefficient TSS separation [10]. Therefore, an alternative treatment system like the upflow anaerobic sludge blanket reactor is required, due to its lesser energy demand, smaller ecological foot print production as well as its overall operation and maintenance cost.

#### 1.1.2. Coagulation–Flocculation and Sedimentation

The addition of coagulant into a reactor vessel containing SWW promotes the formation of large colloidal particles, which are called flocs. The colloidal particles produced in SWW, however, are negatively charged, making them stable and aggregation resistant. Coagulants with positively charged ions are therefore added into the vessel for proper floc formation in order to destabilize the colloidal particles to form flocs and ease the

sedimentation process. Chemicals such as ferric chloride, ferric sulphate, aluminium sulphate, aluminium chlorohydrate, and poly-aluminium chloride were used as coagulants for the SWW treatment. The use of poly-aluminium chloride as reagent showed a total phosphorus (TP), total nitrogen (TN), and COD removals efficiencies of up to 99.9%, 88.8%, and 75.0%, respectively. On the other hand, the sludge volume can be reduced by 41.6% using inorganic coagulants [9,11]. Figure 2 demonstrates coagulation–flocculation and sedimentation processes.

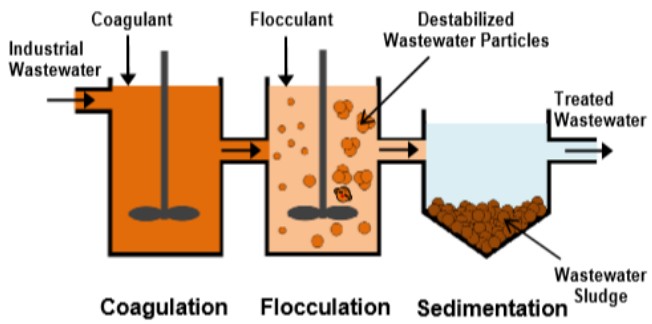

**Figure 2.** Schematic representation of coagulation–flocculation processes [12].

Satyanarayan et al. [13] have reported the use of anionic polyelectrolyte, ferrous sulphate, lime, and alum as coagulants in the treatment of SWW. The results revealed BOD, COD, and TSS removal efficiencies of 38.9, 36.1, and 41.9% using only lime as a coagulant. A significant improvement in COD removal up to 56.8% was realized in the combination of ferrous sulphate with lime. Likewise, an increase in COD removal to 42.6% was recorded in the combination of alum and lime. Tariq et al. [14] investigated the use of alum and lime individually in the treatment of SWW. It was revealed that with the increasing dose of alum, the COD removal reached a maximum of 92% along with high sludge volume, and this rendered the process inefficient. Conversely, 74% COD reduction was realized with an increasing dosage of lime as a single coagulant. Comparatively, the sludge volume generated using lime was quite low compared to that of alum. However, the combination of the two coagulants revealed a maximum COD removal of 85% with a small quantity of sludge volume.

Different contaminants can be removed from the wastewater through coagulation/flocculation which would otherwise be difficult without the application of these chemicals. Limited investment is required for these tanks and dosing units. Nevertheless, the operating costs are a major disadvantage of this strategy. In some situations, significant amounts of coagulant and flocculent are needed to achieve the required level of flocculation. A certain amount of physico–chemical sludge is also produced, which is usually handled externally. These costs may escalate, especially with large volumes of wastewater. The correct dosage of chemicals is also very important for the proper functioning of the process. Therefore, this is not simple for sewage with widely varying composition.

### 1.1.3. Electrocoagulation (EC) Process

Electrocoagulation requires the production of in situ coagulants by electrically dissolving aluminium or iron from aluminium or iron electrodes, respectively. Figure 3 shows the schematic diagram of electrocoagulation processes. Metal ions are produced at the anode and hydrogen gas is emitted from the cathode. Hydrogen gas would also help lift the flocculated particles out of the air. The electrodes can be set in a monopoly or bipolar mode. The products may be made of aluminium or iron in the form of plates or the form of scraps, such as steel turning and milling. The EC process is an advanced treatment technology recently applied to the treatment of SWW. According to Emamjomeh and Sivakumar [15] and Bayar et al. [16], the system is capable of removing pathogens, organics, nutrients, and even heavy metals from SWW by introducing an electric current without the addition of any chemical. Electrodes such as Al, Fe, Pt, $SnO_2$, and $TiO_2$ are commonly utilized for

the EC process, however, Al and Fe are the most widely applied. In the EC process, $M^{3+}$ ions are usually generated on-site with the help of sacrificial anodes. Moreover, studies have shown that these sacrificial electrodes might interact with hydrogen ions in an acidic medium or with an OH- ion in an alkaline medium [17–20]. For instance, the research of Kobya et al. [18] into the EC process treating SWW demonstrated that Al, as an electrode material in the EC process, was responsible for removing up to 93% COD, whereas Fe as an electrode material was able to achieve a maximum of 98% oil and grease removal efficiency. During this process, the influential parameters that lead to the high COD, oil, and grease removal included the pH, operating time, electrode material, and the current density.

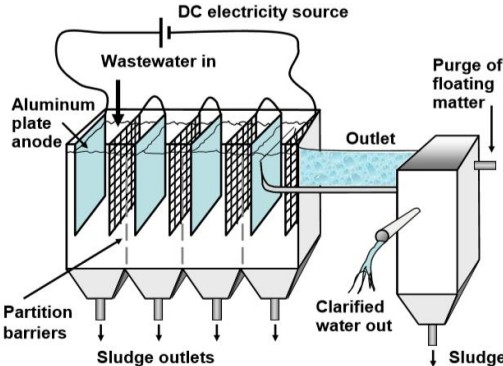

**Figure 3.** Schematic representation of the electrocoagulation processes [21].

An evaluation of the chemical reactions that occur in the process of electrocoagulation reveals that the main electrode reactions (aluminium electrodes) are:

$$\text{Anode: Al} \rightarrow \text{Al}^{3+} \text{ (aq) + 3e} \tag{1}$$

$$\text{Cathode: } 3H_2O + 3e \rightarrow 3/2H_2 + 3O^-$$

The cathode may also be chemically attacked by $HO^-$ ions generated during H2 evolution at high pH [22]:

$$2Al + 6H_2O + 2HO^- \rightarrow 2Al\,(HO)_4{}^- + 3H2$$

$Al^{3+}$ (aq) and $OH^-$ ions generated by electrode reactions (1) react to form various monomeric species such as $Al\,(OH)^{2+}$, $Al\,(HO)_2{}^+$, $Al_2\,(HO)_2{}^{4+}$ and $Al\,(HO)_4{}^-$, and polymeric species such as $Al_6\,(HO)_{15}{}^{3+}$, $Al_7\,(HO)_{17}{}^{4+}$, $Al_8\,(HO)_{20}{}^{4+}$, and $Al_{13}\,O_4\,(HO)_{24}{}^{7+}$, $Al_{13}\,(HO)_{34}{}^{5+}$, which finally transform into $Al\,(OH)_3$ according to complex precipitation kinetics [23].

### 1.1.4. Membrane Technology

Membrane technology is becoming more popular in the treatment of water and wastewater due to regulatory issues towards meeting the stringent water quality requirements. Microfiltration (MF), ultrafiltration (UF), nanofiltration (NF) and reverse osmosis (OS) are the common membrane technologies used for water purification. Figure 4 depicts the different membrane sizes for the treatment of water and wastewater. Depending on the pore size, membranes can remove colloids, particles, and macromolecules. This technology is increasingly applied in the treatment of SWW to remove organic matter and bacteria [24]. The performance of RO in the treatment of secondary effluent of SWW (activated sludge as pre-treatment) was reported by Bohdziewicz and Sroka [25]. The result of parameters like BOD, COD, TN, and TP were found as 50.0, 85.8, 90.0, and 97.5%, respectively. It was concluded that RO was a suitable technique for the post-treatment of SWW effluent. The study of Yordanov [26] on the performance of the UF membrane treating SWW showed

BOD and COD removal efficiencies of around 97.8–97.89 to 94.52–94.74%, whereas TSS and FOG removal were 98 and 99%, respectively.

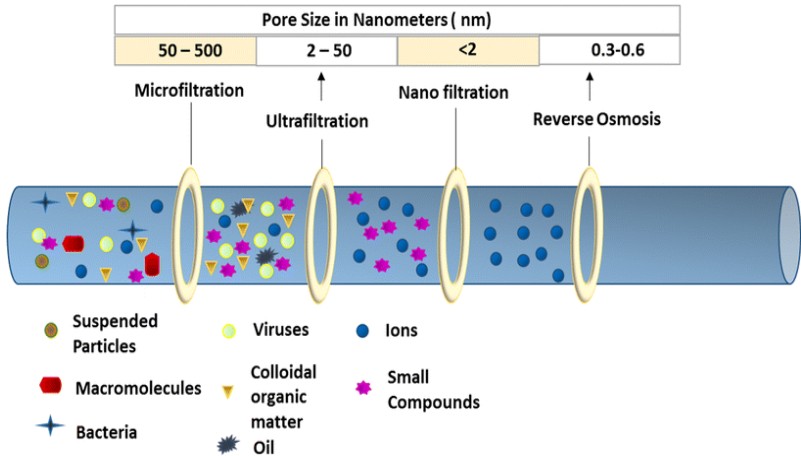

**Figure 4.** Schematic representation of pressure-driven membrane filtration [27].

The investigation of Gürel and Büyükgüngör [28] indicated that a membrane biore-actors could significantly remove nutrients and other organics from SWW. A pilot-scale experiment of anaerobic submerged bioreactor membrane (SAMBR) treating high-strength wastewater (raw tannery wastewater) achieved a higher COD removal efficiency of up to 90% at 6 g/L·day organic loading rate (OLR) and biogas production (0.160 L g COD removed) [29]. The process worked efficiently but was strongly characterized by a high hydraulic retention time (HRT) of 40 h, and as such high energy was spent, although the permeate flux remained at (6.8 LMH) well below the critical flux (17.5 LMH) as established in the earlier work of Hu and Stuckey [30]. Most recently, the filtration performance of an anaerobic membrane bioreactor (AnMBR) treating high strength lipid-rich wastewater and corn-to-ethanol thin stillage was conducted by Dereli et al. [31]. The reactors delivered a high COD removal efficiency of up to 99% under stable operating conditions with an average OLR of 8.3, 7.8 and 6.1 kg COD/m$^3$·day. However, the permeate turned out to be inferior in quality with an increased solid retention time (SRT). Generally, membrane lifetime remains the main concern of investors in the water treatment and wastewater industries. The efficiency of reversing fouling on the membrane surface is being exploited by physical, chemical, and biological methods. Although there were enough physical and chemical methods, the disadvantages are enormous. During aeration, much energy is expended, and sometimes chemicals are used for membrane cleaning, and this activity does not benefit the players in this field in terms of cost and environment.

### 1.1.5. Summary of Physicochemical Treatment Methods

Table 1 summarizes the advantages and disadvantages of the different physicochemical treatment methods of slaughterhouse wastewater.

**Table 1.** Advantages and disadvantages of physicochemical methods.

| Methods | Advantage | Disadvantage |
|---|---|---|
| Dissolved air floatation | ■ It can achieve 30–90% COD and 70–80% BOD removal efficiencies.<br>■ Moderate to high nutrient removal.<br>■ Tends to carry fairly solid binding particles and can easily be skimmed from the water's surface. | ■ High energy demand due to aeration.<br>■ Chemical addition which renders the sludge unsuitable as fertilizer.<br>■ Inefficient total suspended solid separation.<br>■ Lacks energy recovery facilities.<br>■ Frequent system failure.<br>■ High cost of operation and maintenance. |

**Table 1.** *Cont.*

| Methods | Advantage | Disadvantage |
|---|---|---|
| Coagulation–flocculation and sedimentation | ■ Promotes large colloid formation which can easily sediment.<br>■ TP, TN, and COD removals efficiencies of up to 99.9%, 88.8%, and 75.0% can be achieved. | ■ Huge quantity of chemical is applied.<br>■ Large volume of sludge is generated causing an additional cost of treatment.<br>■ Difficult to handle or reuse.<br>■ Landfill disposal or incineration is usually the only option available to handle the sludge.<br>■ Lacks energy generating facilities. |
| Electrocoagulation (EC) process | ■ The system is capable of removing pathogens, organics and other nutrients by introducing electric current.<br>■ High COD and FOG removal efficiency (>90%). | ■ High energy demand and not cost effective.<br>■ Lack energy generation facilities especially in the treatment of organic wastewater to high energy potentials. |
| Membrane technology | ■ Depending on the type of membrane, the technology is capable of achieving 97.8–97.89% and 94.52–94.74% BOD and COD removal efficiencies in the treatment of slaughterhouse wastewater. | ■ Characterized by frequent fouling.<br>■ During aeration, much energy is expended, and sometimes chemicals are used for membrane cleaning.<br>■ High energy input and zero energy output especially in the treatment of slaughterhouse wastewater. |

## 2. Biological Treatment

In the treatment of SWW, biological treatment is applied as a secondary treatment to reduce the concentration of BOD and other soluble compounds after primary treatment [32]. Depending on the characteristics of SWW, the biological process is applied when aerobic and anaerobic digestion are operating individually or as combined systems with packing material [33]. Unlike the physicochemical process, the biological treatment process employs the use of microorganisms to remove organics from SWW effluent. Mittal [9] demonstrated that the biological method that properly applies the aerobic or anaerobic process could remove about 90% BOD from SWW effluent. There exist different biological systems, which include anaerobic, aerobic, facultative lagoons, the activated sludge process and trickling filters [34]. Generally, the mechanisms of biological treatment are a function of bacterial consortium to break down organic waste.

### 2.1. Anaerobic Treatment

Anaerobic treatment technology has proven to be a vital area of research in the management of organic waste. This is because the technology tends to offset the setbacks exhibited by aerobic and physicochemical methods [35]. Considering the portion of the industry's waste and its by-products that have recovery potential for direct reuse, including nutrients and methane gas, anaerobic systems are a suitable technology for handling high-strength industrial wastewater such as swine and SWW. It is seen in the discharged effluent consistency, material recovery, energy generation, and sludge output, handling, and storage [36]. The biogas composition consists of methane (55–70%) and carbon dioxide (30–45%) under strictly anaerobic conditions. Other contaminants are nitrogen (0–15%), oxygen (0–3%), water (1–5%), hydrocarbons (0–200 mg m$^{-3}$), ammonia (0–100 ppmv) and siloxane (0–41 mg Si m$^{-3}$) [37]. Typical anaerobic digestion systems include anaerobic lagoon (AL), anaerobic filter (AF), anaerobic baffled reactor (ABR), and upflow anaerobic sludge blanket reactor (UASB).

#### 2.1.1. Anaerobic Lagoon

Anaerobic lagoons (ALs) have been widely applied in the degradation of wastewater, especially in developing countries with hot weather. The method used largely depends on the climate, location, availability of land, and proximity to urban areas [9]. The influent is

usually introduced through the bottom of the system and is not mechanical mixed. A layer of scum frequently forms on the surface of the lagoon, ensuring the system is confined to anaerobic conditions with low heat loss. Figure 5 showed a typical anaerobic lagoon. According to the literature [38,39], the COD, BOD, and TSS removal efficiency of a typical AL with a depth of 3–5 m and a hydraulic retention time of 5–10 days were found as 96%, 97% and 95%, respectively.

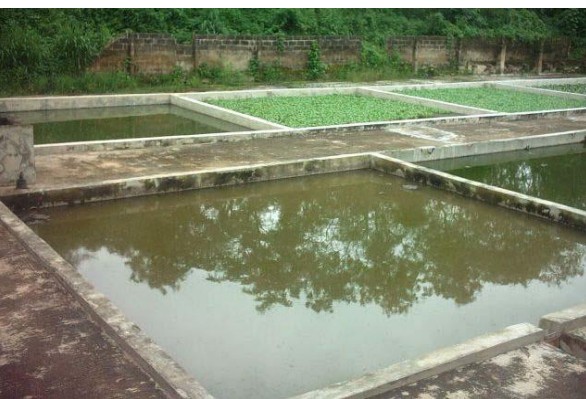

**Figure 5.** Anaerobic lagoon for wastewater treatment [40].

However, this system's pitfalls include odour generation and weather dependency, coupled with a long HRT and requiring a large area of land to operate. Thus, to reduce odour and smell, the synthetic floating cover is normally employed to collect biogas and trap the odour, as shown in Figure 6.

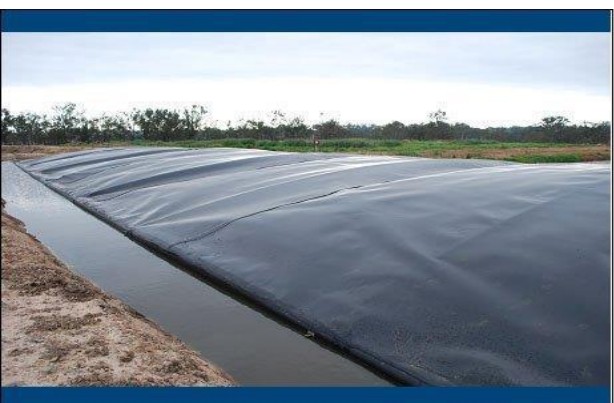

**Figure 6.** Anaerobic lagoon with cover [41].

Moreover, the synthetic cover must be durable and able to resist change in temperature, or ice and snow accumulation [9]. ALs are frequently the preferred method of treating SWW due to their simplicity as well as their low operational and maintenance costs, especially in developing countries [39].

2.1.2. Anaerobic Filters

Anaerobic filters are usually run in upflow mode, as the system has a lower risk of washing out the fixed biomass, as shown in Figure 7.

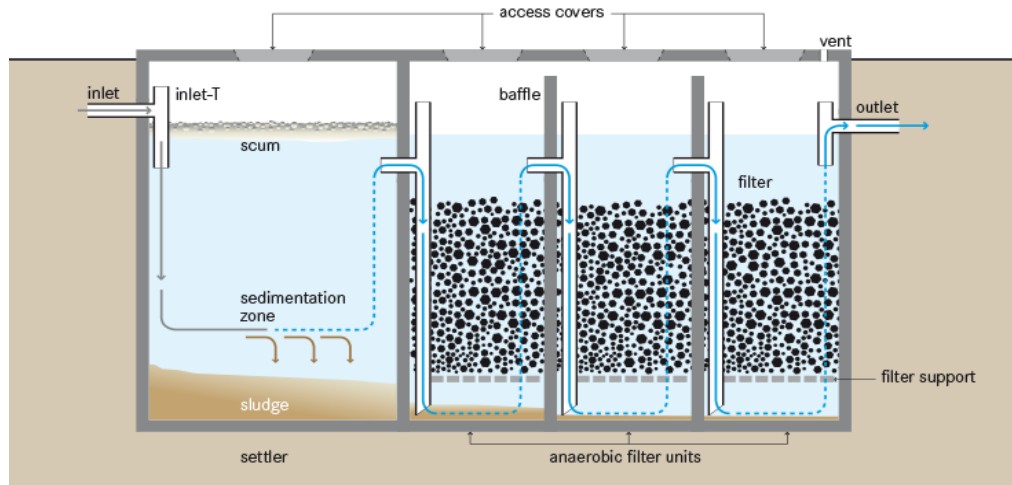

**Figure 7.** Anaerobic filter for wastewater treatment [42].

In order to ensure an even flow regime, the water level must cover the filter media by at least 0.3 m. Hydraulic retention time (HRT) is the most important design parameter to influence filter efficiency. For bacteria to grow, the ideal filter should have a large area, with pores small enough to avoid clogging. The surface area ensures increased contact which ultimately degrades it between the organic matter and the attached biomass. Ideally, the material must occupy a surface area of 90 to 300 $m^2$ per $m^3$ of the volume of the reactor. The typical filter content sizes vary from 12 to 55 mm in diameter. Widely used products include dirt, crushed stones or bricks, cinder, pumice, and specially shaped plastic parts, depending on local availability. The systems are used for the secondary treatment of SWW to achieve high solids removal and biogas production. These systems usually work in series and have a fixed bed biological reactor coupled with a filtration chamber. When the SWW flows through the filtration chambers, large and medium suspended particles are confined within; then, the active biomass attached to the surface of the filter degrades the particulate organic matter [9]. Gannoun et al. [43] examined the performance of upflow anaerobic filters (UAFs) treating SWW at mesophilic and thermophilic temperatures. The results showed that at an organic loading rate (OLR) of 9 g/L/d, the COD removal efficiency was 90% at mesophilic, and only 72% was achieved at the thermophilic condition. On the other hand, the mesophilic (35°C) treatment of SWW at a high organic loading rate of OLR 10.05 kg/$m^3$day and HRT of 12 h was evaluated by Rajakumar et al. [44]. The system recorded a COD removal rate of 79% with a varied methane production between 46 and 56% on the average. The experiment of Stets et al. [45] evaluated the influence of substrate characteristics, microorganisms present in the sludge, and the supporting media in AF. The results showed a maximum COD and TN removal efficiency of 80 and 90% at an HRT of 1 day. The major drawback of anaerobic baffle reactor is the need for relatively higher temperatures for optimum service, but this is not an obstacle in tropical countries.

### 2.1.3. Anaerobic Baffled Reactor

Anaerobic baffled reactors (ABRs) consist of a series of compartments with inlet and outlet, in which SWW flows in from beneath and above. The reactor is commonly referred to as an optimized version of a septic tank, and the diagrammatic representation of the reactor design and its characteristic dimensions is shown Figure 8.

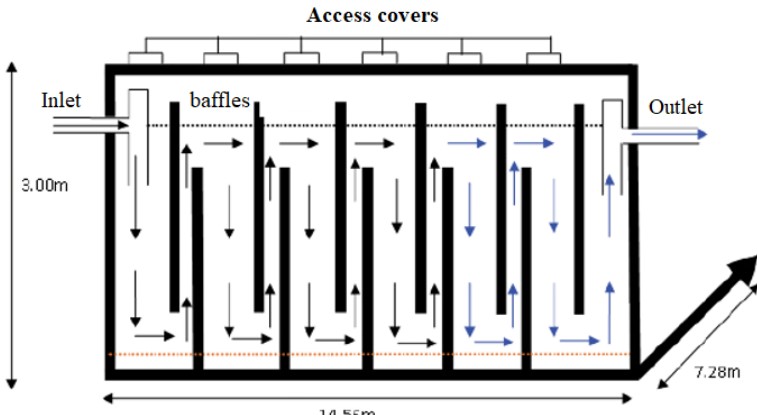

**Figure 8.** Anaerobic baffled reactor [46].

The purpose of using the anaerobic baffled reactor is to provide the enhanced removal and digestion of organic matter as well as of microorganisms present in the influent. The design objective was to increase the contact time between the suspended or dissolved contaminants and biomass and to minimize the amount of sludge washout in the ABR effluent. This can be achieved by maximizing the hydraulic retention time, the number of passes through the sludge bed (i.e., the number of compartments), and the upflow rate to reduce the transport of solids within processing and capital cost constraints as determined by solid retention. Two six-compartment anaerobic baffled reactors to be installed in series are usually designed to achieve maximum treatment rates. This engineered two six-compartment ABR offers 96 h (48 h for each ABR) hydraulic retention period which by far was higher than the 48–92 h ranges for high peak-flow output levels and the 20–60 h, which allowed high-performance treatment for domestic wastewater. The peak up-flow rate of 0.54 m/h was proposed by Foxon and Buckley [47], and peak flow factor of 1.8 resulted in an upflow rate of 0.30 m/h. This value corresponds to the one suggested by Tilley et al. [42], which is <0.6 m/h. The study of Kuşçu and Sponza [48] revealed that a significant improvement in COD and BOD removals up to 90% was achieved in the upflow compartment. A laboratory-scale study of the performance of combine ABR and UV/$H_2O_2$ treating SWW with a total organic carbon (TOC) concentration of 973 mg/L exhibited high organic carbon removal efficiency up to 95% [49]. One major drawback of this type of reactor is that the system does not have auxiliary mechanisms for the retention of biomass, in the case of large variations and extreme peaks of the influential flow.

### 2.1.4. Upflow Anaerobic Sludge Blanket Reactor

The development of the UASB technology dated back to the late 1970s, and was initially developed for the anaerobic treatment of liquid waste streams with a high concentration of COD (1.0 to 200 g COD $L^{-1}$) and low solid content [50,51]. The upflow anaerobic sludge blanket (UASB) reactor is a tank with a sludge bed occupying half or less the volume of the total tank from the bottom of the tank. UASB reactor consists of three zones: the sludge zone containing the biomass, substrate-like SWW, and the gas zone above the substrate [52,53]. As the name implies, upflow, the SWW enters from the bottom and flows upward with a high or low velocity through the sludge blanket, which then exits from the top as an effluent as illustrated in Figure 9. Depending on the prevailing parameters, literature have reported a satisfactory performance of the UASB reactor in degrading SWW.

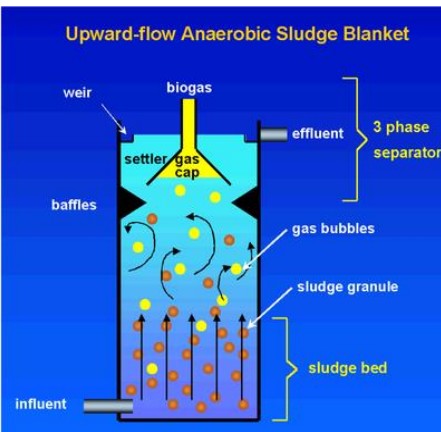

**Figure 9.** Conceptual diagram of upflow anaerobic sludge blanket (UASB) [54].

　　　The many advantages of UASB reactors include less sludge production, energy recovery, and the overall low cost of application [55]. Moreover, the bacteria can withstand a long period of starvation without dying, and only one discharge of sludge is required per year for a UASB reactor with around 4 m high. Tropical countries stand to benefit more in the use of the UASB reactor because they work better at mesophilic conditions. The research of Caldera et al. [56] demonstrated that a high COD removal efficiency of up to 94.31% from a UASB reactor treating SWW under mesophilic condition. The substrate concentration varied from 1820 to 12,790 mg/L, and the experiment lasted for 90 days at HRT of 24 h. In another development, Chávez et al. [57] reported the 95% BOD removal efficiency of UASB treating slaughterhouse waste at an optimum OLR 31,000 mg/L under mesophilic conditions at HRT 3.5 and 4.5 h. The work of Miranda on the 800 m$^3$ full-scale UASB reactor treating SWW with an influent of COD concentration in the range of 1400–3600 mg/L and oil and grease content between 413 and 645 mg/L, respectively. The results of their experiment revealed that around 70–92% COD and 27–58% oil and grease removal efficiencies. Moreover, an optimum COD removal efficiency of 90% was also revealed in the study of Rajakumar and Meenambal, [58] at an HRT of 10 h, varying the COD concentration from 3000 to 4800 mg/L in the UASB reactor. Mijalova et al. [59] analysed the output of a UASB reactor treating SWW after solid separation under the ambient condition. It was reported that the efficiencies of COD removal improved in relation to OLRs. With an influent COD concentration of 3437 mg/L, the system recorded a high COD removal efficiency of 90%. While UASB reactors are found to be effective for SWW treatment, compliance with current water quality standards for water body discharge requires a post-treatment process. Table 2 shows the review of the performance of previous works on UASB reactors treating SWW and other wastewater. However, the system shortfall of sludge washout at elevated upflow velocity and the slow-growing methanogenic bacteria. The performance of various UASB reactors treating slaughterhouse and other wastewaters are shown in Tables 2 and 3.

**Table 2.** The performance comparison of different UASB reactor treating slaughterhouse wastewater.

| Type of Substrate | OLR | HRT (h) | Temperature (°C) | %COD Removal | Biogas Production | SMP (L g$^{-1}$ COD$_{added}$) | Scale | Reference |
|---|---|---|---|---|---|---|---|---|
| Slaughterhouse wastewater | 0.2–1.4 kg COD/m$^3$d$^{-1}$ | 12 | 24–35 °C | 30–62% | 3.45 L/d | NR | Lab | [60] |
| Slaughterhouse wastewater | 1.46 to 2.43 kg COD/m$^3$d$^{-1}$ | 18–27 | 25 °C | 70–92% | NR | NR | Full | [61] |
| Slaughterhouse wastewater | 0.64 - 2.95 kg COD/m$^3$d$^{-1}$ | NR | 35 °C | 58.4% | 270 mL/d | NR | Lab | [62] |
| Slaughterhouse wastewater | 13–39 kg $_S$COD/m$^3$d$^{-1}$ | 2–7 | 33 °C | 75–90% | NR | 200–280 LCH$_4$/kg SCOD$_{removed}$ | Pilot | [63] |
| Slaughterhouse wastewater | 4–15 kg COD/m$^3$d$^{-1}$ | 0.88, 0.71 0.44, 0.30 | 20.9–25 °C | 90% | 0.020 ± 70.013 0.039 ± 70.010 0.095 ± 70.008 m$^3$/d | 0.239 ± 70.095 0.266 ± 70.005 m$^3$/kg COD$_{removed}$ | Lab | [59] |
| Poultry slaughterhouse wastewater | 2.1 kg COD/m$^3$d$^{-1}$ | 1–5 | NR | >80% | NR | NR | Lab | [64] |
| slaughterhouse wastewater | 1.27–17 kg COD/m$^3$d$^{-1}$ | 4–0.3 | 35 °C | NR | 0.680–3.790 L.L$^{-1}$. day$^{-1}$ | NR | Lab | [65] |
| Poultry slaughterhouse wastewater | 15 kg COD/m$^3$d$^{-1}$ | 24, 16, 12, 10 and 8 | 29–35 °C | 78% | 20.3 L/d | 0.24 m$^3$CH$_4$/ kg COD$_{removed}$ | Lab | [44] |
| Slaughterhouse wastewater | 0.32, 0.51, 1.16 and 2.31 kg COD/m$^3$d$^{-1}$ | 22, 14, 6 and 3 | 29.6 ± 1.40 °C | 43.39–84.54% | 143.9 m$^3$ | 0.09 ± 0.03 to 0.22 ± 0.02 m$^3$/ kgCOD$_{removed}$ | Pilot | [66] |
| Slaughterhouse wastewater | 1.2 kg COD m$^{-3}$ d$^{-1}$ | 24 | 30 ± 1 °C | 70% | NR | NR | Lab | [67] |
| Slaughterhouse wastewater | 0.54 | 15.6 | 35 ± 1 °C | 50.9 | NR | 100 mL CH$_4$/gCOD$_{added}$ | Lab | [68] |

**Table 3.** UASB performances on various types of wastewater.

| Type of Substrate | Temperature | Influent COD | HRT (h) | OLR (d) | Biogas Produced | COD Removal (%) | References |
|---|---|---|---|---|---|---|---|
| Low strength wastewater | Ambient temperature (20–35 °C) | 500 mg/L | 3 | 4 kg COD/m$^3$/d | 141 L/kg COD $_{removed}$ | 90–92% | [69] |
| Domestic wastewater | Ambient temperature | - | 7.6 | 1.21 kg COD/m$^3$/d | 0.34 m$^3$CH$_4$/ g COD $_{removed}$ | 85% | [70] |
| Wheat straw stillage | 55 °C | 70 g/L | 48 | 17.1 g COD/L/d | 154.8 mL CH$_4$/g COD | 76% | [71] |
| Composite chemical wastewater | 29 ± 2 °C | 6600 mg/L | 37 | 4.25 kg COD/m$^3$/d | 0.3 m$^3$CH$_4$/kg COD $_{removed}$ | 62% | [72] |
| Potato leachate wastewater | 37 °C | 20 g/L | | 6.1 g COD/L/d | 0.23 L CH$_4$/ g COD $_{degraded}$ | 93 ± 5.3% | [73] |
| Seaweed leachate | 37 ± 1 °C | 7.3 ± 1.1 g/L | 88.8 | 2.9 g COD/L/d | 0.23 ± 0.03 NL CH$_4$/g COD$_{added}$ | - | [74] |
| High-strength municipal wastewater | 30 °C | 1200 mg/L | 4 | 7.2 kg COD/m$^3$/day | 306.6 mL CH$_4$/g COD $_{removed}$ | 85% | [75] |
| Potato juice | 37 °C | 25.2 g/L | 240 | 2.5 g COD/L/d | 250 ± 6 mL CH$_4$/ gVS $_{added}$ | - | [76] |
| High salinity wastewater from heavy oil production | 30 ± 2 °C | 350–640 mg/L | 48 | 0.23 kg COD/m$^3$/d | - | 65.08% | [77] |
| Low strength wastewater | Ambient temperature (24–35 °C) | 700–1000 mg/L | - | 1.293 kg COD/m$^3$/d | 457 L/kg COD $_{removal}$ | 90.8% | [78] |

A literature review on the anaerobic digestion process was carried out to identify the main concepts and operating parameters associated with the upflow anaerobic sludge blanket reactors, as the selected technology and the issues relevant to the investigations (Tables 2 and 3). The review summarizes the main findings related to COD removal efficiencies and the biogas production of the UASB reactors treating slaughterhouse and slaughterhouse-related wastewater. A considerable amount of attention was given to the effects of OLR and HRT on the efficiency of the systems which are potentially the key parameters for the digestion of slaughterhouse and other organic wastewater. A laboratory scale study of the anaerobic digestion of slaughterhouse wastewater by Chollom et al. [60] showed 30–62% COD removal efficiency. It can be seen that, the COD removal efficiency and the biogas production were low. These could be due the low temperature and the HRT. Similarly, Batubara et al. [62], Nacheva et al. [59] and Del Nery et al. [64] studied the anaerobic digestion of slaughterhouse wastewater at the laboratory scale. However, it was observed that the systems were highly characterized by long HRT and low biogas production. On the other hand, a pilot scale investigation showed a high COD removal efficiency [63,66]. Conversely, the work of Worku [62] revealed a very low specific methane production (SMP), and this could be attributed to the low COD of the effluent (COD = 7049.07 ± 306.42 mg/L).

The application of the UASB reactor achieved considerable success in the treatment of a wide range of other organic industrial effluents including low- and high-strength domestic wastewater as depicted in Table 3. However, the systems were characterized by long HRT, although the study of Singh et al. [69] and Hazrati and Shayegan [75], showed a lower HRT and high COD removal efficiency, but the influent COD concentration was very low as compared to other wastewater presented in Table 3. Therefore, there is the need to modify the UASB reactor to treat high strength wastewater at a higher OLR and short HRT and to also comply with the stringent environment regulations.

### 2.1.5. Suspended and Attached Growth Process

The waste flows through and around free-floating microorganisms in a suspended-growth system, such as activated sludge processes (Figure 10A,B) (also aerated lagoons and aerobic digestion), accumulating into biological flocks which then settle out of the wastewater. the influent wastewater characteristics such as (COD), total N, and total P and operating parameters like HRT, SRT, dissolved oxygen (DO), return activated sludge (RAS), and mixed liquor recirculation (MLR) flow rate have significant impacts on the performance of anaerobic, anoxic and aerobic system (Figure 10A). However, the competition for organic substrates among the bacterial population, there is a concern about the adverse effects of the returning sludge on the growth of the bacteria, which prefer to grow under alternating anaerobic and aerobic conditions. Therefore, to overcome the inherent drawbacks of the anaerobic, anoxic and aerobic processes, a reverse anoxic, anaerobic and aerobic process (Figure 10B) is believe to improve the performance of the bacteria through aerobic conditions.

The microbial mass is retained as flocs in suspended growth systems in the mixed liquor of the reactor. Mechanical mixers or gas injection hold these flocs in suspension with agitation. The air in aerobic processes and biogas in anaerobic systems will normally be the latter. Agitation facilitates intimate interaction between the substrate and the biomass. Microbes are attached to the support medium in a thin layer in biofilm systems. The latter can be a static bed or a moving bed. Fixed or stationary beds are usually moulded plastic or gravel shapes, while moving beds may include granular activated carbon or sand grains. These beds of support medium may be submerged in a mixed liqueur during the operation of the reactor or otherwise exposed to the air and wastewater. The bulk of the aerobic systems used in wastewater treatment plants are suspended growth systems. Other examples of suspended growth systems include the aerated lagoon, oxidation ditch, and batch sequencing reactor. However, biofilms with suspended growth can sometimes be integrated into such aerobic systems. Attached growth systems (also known as fixed-

film processes) are processes for the treatment of biological wastewater with the biomass attached to certain types of media. Figure 11a shows the laboratory scale attached growth system, while Figure 11b represents a pilot-scale attached growth system, likewise.

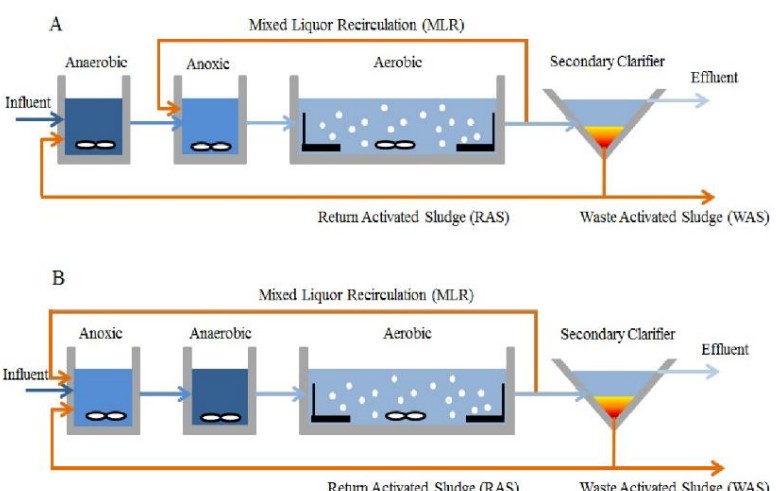

**Figure 10.** Schematic of two conventional activated sludge processes [79]. It's the same diagram: (**A**) and (**B**). The only difference is that the mixed liquor recirculation (MLR) is returned to aerobic condition (**A**) and sometimes is to return anoxic (**B**).

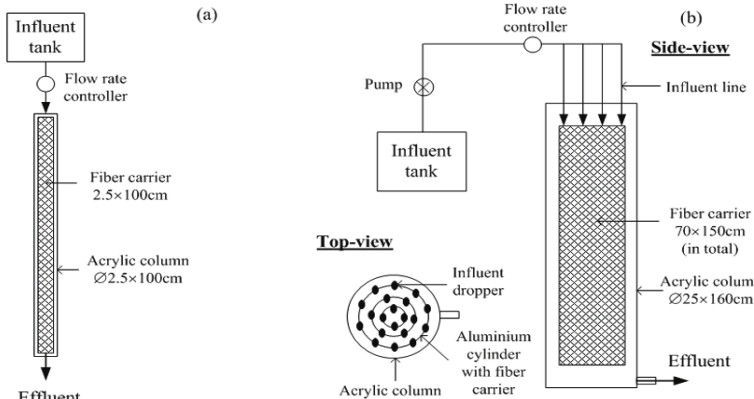

**Figure 11.** A schematic diagram of attached growth system [80]. Figure represents attached growth system in (**a**) lab-scale and (**b**) pilot-scale.

Figure 12 shows the difference between the suspended and the attached growths. The growth formed in the media is a mixture of mainly aerobic microorganisms.

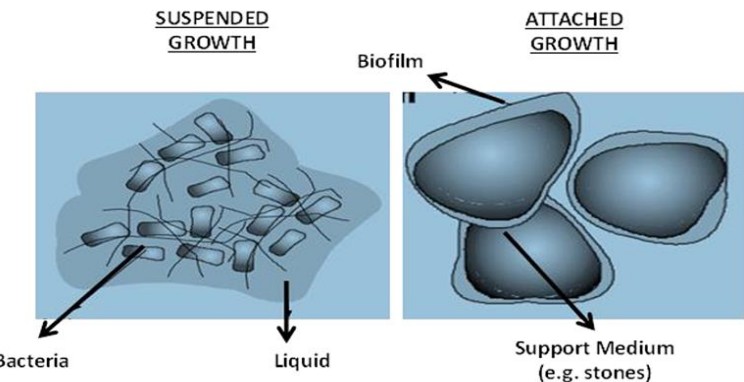

**Figure 12.** Typical examples of suspended and attached growth system [81].

These are similar to those found in other secondary biological treatment processes. The microorganisms include ciliates, rotifers, nematodes, and many others that are free-swimming and stalking. Attached growth processes are easy to operate and resistant to shock loads but are less versatile than activated sludge processes for process control.

### 2.1.6. Summary of Biological Treatment Methods

Table 4 summarizes the advantages and disadvantages of the different physicochemical treatment methods of slaughterhouse wastewater.

**Table 4.** Advantages and disadvantages of biological treatment methods.

| Methods | Advantage | Disadvantage |
|---|---|---|
| Anaerobic lagoon (AL) | ■ It is the most preferred method of treating slaughterhouse wastewater (SWW) due to its simplicity, low cost of operation and maintenance.<br>■ No mechanical mixing is required.<br>■ A typical AL with a depth of 3–5 m and a hydraulic retention time (HRT) of 5–10 days usually achieve COD, BOD, and TSS removal efficiency of up to 96%, 97% and 95%.<br>■ Odour and smell reduction is achieved by employing synthetic floating cover to collect biogas and trap the odour. | ■ Depends largely on the climate, location, availability of land, and proximity to urban areas.<br>■ A layer of scum frequently forms on the surface.<br>■ Long HRT.<br>■ The synthetic cover may experience leakages with time due to fluctuation in temperature. |
| Anaerobic filters (AF) | ■ Have 3 chambers with filters and the active biomass attached to the surface of the filter degrades the particulate organic matter.<br>■ Runs in upflow mode, hence mechanical mixing is not required. | ■ Relatively high temperature is required.<br>■ Sludge sedimentation.<br>■ The systems are used for secondary treatment of SWW to achieve high solids removal and biogas production.<br>■ The filters can easily clog. |
| Anaerobic baffled reactor (ABF) | ■ Increases the contact time between suspended or dissolved contaminants and biomass and minimizes the amount of sludge washout. | ■ Lacks biomass retention mechanism.<br>■ Long HRT.<br>■ Biomass could easily washout at peak flow and short HRT. |
| Upflow anaerobic sludge blanket reactor | ■ Less sludge production.<br>■ Energy and materials recovery.<br>■ Can operate at higher organic loading rate (OLR).<br>■ Required small reactor volume and space for installation.<br>■ Low operation and maintenance cost. | ■ Long start-up period due to slow growing methanogenic bacteria.<br>■ Sludge washout at low HRT.<br>■ Scum formation on the substrate surface.<br>■ Effluent post treatment. |
| Suspended growth (activated sludge process) | ■ Commonly applied method for a large volume of wastewater.<br>■ Biomass recirculation simplified the continuous operation.<br>■ Good effluent quality. | ■ Mainly aerobic bacteria, hence no energy is produced.<br>■ Require large space.<br>■ High cost of operation and maintenance.<br>■ Not flexible (in case of change in waste concentration).<br>■ Sludge disposal is required on large scale. |
| Attached growth process | ■ The attached growth processes are low maintenance, low energy requirements, and, overall, less technology involved.<br>■ Very effective for biochemical oxygen demand (BOD) removal, nitrification, and denitrification | ■ Mostly suitable for the treatment of wastewater for small communities.<br>■ Larger area requirement, ineffective in cold weather, and creates odour problems. |

*2.2. Concluding Remarks*

The literature review revealed that anaerobic digestion appeared as a promising technology for the treatment of low- and high-strength slaughterhouse wastewater, although it is a complex and sensitive process. The operation of the anaerobic reactor is highly dependent on the temperature, pH, hydraulic retention time, and loading rates as well as wastewater and biomass characteristics. It was also found that, for good substrate degradation, anaerobic reactors' optimal temperature conditions include psychrophilic (<25 °C), mesophilic (25–40 °C) and thermophilic (>45 °C). However, most of the studies showed that the reactor performance was more stable at mesophilic condition (25–40 °C). Likewise, stable pH condition usually exists between 6.5 and 7.5. Among the various anaerobic reactors, UASB reactor showed high biogas production and COD removal efficiency at high OLR. The system is also characterized by low sludge production compared to other physicochemical treatment methods, where less energy is applied and high energy is generated and the overall cost of operation and maintenance is lower. While conventional UASB reactors appear to be a promising choice for the recovery of energy from organic wastewater, including slaughterhouse wastewater (SWW), there are potential problems associated with the reactor. These include the long start-up period due to the slowly growing microorganism, sludge washout at low HRT, scum formation on the substrate surface—especially in in the treatment of SWW—and the suspended solid accumulation at high inflow velocity with low HRT.

Several researchers have studied the role of attached growth media in increasing the concentration of the microbial population. However, most of the systems were highly characterized by long HRT and low OLR especially in conventional UASB reactors (Table 2). Furthermore, most of the media used for microbial attachment in anaerobic reactors are usually made of plastic materials with a smooth surface and normally configured in suspended moving media, which leads to the overflow of the media on the substrate surface during influent pumping at higher velocity and these consequently result in insufficient contact between the microbes and the media. Additionally, as a result of the separation between microbes and the media, the microbes could easily washout during effluent discharge. Despite the numerous studies on this subject, none of the previous research has focused on the comparison between the performance of the conventional UASB reactor and UASB reactor with fixed attached growth media with a rough and large surface area that confines the whole sludge zone in a UASB reactor treating high strength cattle slaughterhouse wastewater (CSWW). The literature review has thus identified some key gaps in the knowledge, especially in aerobic and anaerobic treatment processes, and indicated a number of concepts and tools that may be useful in future research. For instance, aerobic processes are highly characterized by high energy demand, the large area of land for installation, huge quantity of sludge production and inefficient small and medium scale industries. Similarly, most anaerobic systems required a long HRT for bacterial growth, and sludge easily washes out along with the large microbial population during effluent discharge and is highly temperature dependent. Therefore, further research on the use of organic or inorganic waste materials or cellulose materials should be conducted to further harness the most cost-effective methods of slaughterhouse wastewater treatment.

**Author Contributions:** Conceptualization, M.A.M. and S.I.; methodology, M.A.M.; validation, S.I.; formal analysis, M.A.M.; investigation, M.A.M.; resources, S.I.; data curation, S.I.; writing—original draft preparation, M.A.M.; writing—review and editing, S.I.; visualization, S.I.; supervision, S.I. All authors have read and agreed to the published version of the manuscript.

**Funding:** This research was financially supported by the Ministry of Higher Education Malaysia through Fundamental Research Grant Scheme (FRGS/2/2014/TK02/UPM/02/6) and Tenaga Nasional Berhad Research Sdn. Bhd. through Industrial grant (TNBR/Biogas/2019/UPM/6380035).

**Institutional Review Board Statement:** Not applicable.

**Informed Consent Statement:** Not applicable.

**Data Availability Statement:** Not applicable.

**Acknowledgments:** The authors acknowledge the support received from Ministry of Higher Education Malaysia. Also, Tenaga Nasional Berhad Research Sdn. Bhd and the Universiti Putra Malaysia, for the preparation, execution, and writing of this article.

**Conflicts of Interest:** The authors declare no conflict of interest.

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
