# Peer review of "Physical and Biological Treatment Technologies of Slaughterhouse Wastewater: A Review"

_sustainability, doi:10.3390/su13094656_

Round 1

Reviewer 1 Report

Dear Authors,

I attach comments in a pdf file, in the form of comments. The manuscript contains many minor errors, please carefully review and correct the text.

Yours faithfully, 

Rev

Author Response

Thank you very much for your knowledge impacting comments and suggestions. The authors greatly appreciate your effort. We have made the necessary corrections as suggested. 

Thank you

Reviewer 2 Report

After going through the manuscript " Physical and Biological Treatment Technologies of Slaughterhouse Wastewater: A Review", I would give my comments below.

- I think it's a parallel work with some new review papers that publish in recent months such as:

"Bustillo-Lecompte, C. F., & Mehrvar, M. (2015). Slaughterhouse wastewater characteristics, treatment, and management in the meat processing industry: A review on trends and advances. Journal of environmental management161, 287-302.",

“Bustillo-Lecompte, C., & Mehrvar, M. (2017). Slaughterhouse wastewater: treatment, management and resource recovery. Physico-chemical wastewater treatment and resource recovery, 153-174.”
and
"Fard, M. B., Mirbagheri, S. A., Pendashteh, A., & Alavi, J. (2019). Biological treatment of slaughterhouse wastewater: kinetic modeling and prediction of effluent. Journal of Environmental Health Science and Engineering17(2), 731-741."
There is not a new review manuscript for the present, so, what makes this review different from the others and from the most recent ones?

- Abstract should be rewritten. The general information should be concisely. Instead, more details of the reviewed aspects should be presented.

- Table 1 and 4 need new rows about the characterization of methods used and more details.

- Should be provided a comprehensive part between all of the treatment of Slaughterhouse wastewater in the experimental and field-scale till now used. Add another table or tables.

- A review paper not only should summarize recently published works, but also should contain critical and comprehensive discussions. Therefore, check writing for the whole manuscript. The review should not be presented by listing what have done by others.
- Technical terms are misused through the manuscript and the writing needs a revision.

- Section of drawbacks and future could be increased quality of the manuscript.

- “Biological Treatment” is written simply, most recent research and innovation in Cellulose Polyethyleneimine performances should be reviewed to show the gap of knowledge. This part should be extended with recently research papers.

Author Response

The authors sincerely appreciate your kind and knowledge impacting comments and suggestions. Please find attached a copy of our response to the issues raised.

Thank you

Round 2

Reviewer 2 Report

The article can be accepted for publication.

Author Response

The authors sincerely appreciate your knowledge impacting comment and suggestion. The gap in knowledge has added in lines 97 to 104 for your consideration.

Thank you
